# The Efficacy of Hyaluronic Acid Binding (HAB) in the Treatment of Male Infertility: A Systematic Review of the Literature

**Róisín Ní Dhuifin [1], Darren K. Griffin [2]** and **Therishnee Moodley [3],***

[1] School of Medicine, University of Dundee, Dundee DD1 9SY, UK; roisin.dhuifin@tfp-fertility.com
[2] School of Biosciences, University of Kent, Canterbury CT2 7NF, UK; d.k.griffin@kent.ac.uk
[3] Centre for Reproductive Medicine, St. Bartholomew's Hospital, London EC1A 7BE, UK
* Correspondence: therish.moodley@nhs.net

**Abstract:** Hyaluronic acid (HA)-binding is reported to predict the fertilising capacity of spermatozoa, while HA-bound sperm selection is reported to reduce the incidence of miscarriage. However, the clinical effectiveness of these techniques remains uncertain. This work investigated the prognostic value of sperm-HA binding (HAB) as a predictor of treatment outcomes, and whether HAB-sperm selection for Invitro fertilisation (IVF)/intracytoplasmic sperm injection (ICSI) improves clinical outcomes or reduces miscarriage rates. A systematic review of the literature was carried out. A modified version of the Downs and Black Checklist was used to assess bias and study quality on eleven selected studies. No significant correlations were found between HAB score and fertilisation, clinical pregnancy, or live birth rates (low-quality evidence). Three studies reported a significant reduction in the incidence of miscarriage, including a Cochrane review (low-quality evidence). While the prognostic value of HAB scores is currently undetermined, there is evidence that HAB-sperm selection prior to insemination reduces the incidence of miscarriage following ART. Moreover, there are no reports of detrimental effects of HAB-sperm selection on treatment outcomes when compared with conventional IVF or ICSI. Therefore, it is unclear why it is assigned as a treatment "add-on" with a red light by the HFEA, and why its routine use is not recommended.

**Keywords:** hyaluronan; HAB-select; physiologic ICSI (PICSI)



## 1. Introduction

Technological advances for the treatment of infertility have been considerable over the last decade. Despite this, in vitro fertilisation (IVF) success rates remain typically under 50% in most clinics [1]. While factors such as embryo culture conditions may contribute to this, gamete quality also plays a key role in embryonic development [2,3].

It has been reported that the integrity of the paternal genome can significantly impact successful embryo development and subsequent pregnancy outcomes [4–6]. Patients suffering from male factor infertility commonly present with higher frequencies of DNA damage and packaging defects. Consequently, in cases of intracytoplasmic sperm injection (ICSI), there may be an increased risk of selecting sperm of suboptimal fertilising potential [6]. Moreover, for female patients over 35 years of age, the DNA repair mechanisms of the oocyte may be compromised and unable to overcome significant sperm DNA damage. A combination of these factors may account for poor success rates, as well as the increased risks of pregnancy loss associated with assisted reproductive technology (ART) cycles when compared to natural conception [6].

Male infertility is gradually on the rise [7], however sperm preparation and semen analysis have remained largely unchanged since the inception of IVF. Crucially, and the fertilising potential of individual sperm is not assessed [8], limiting diagnostic and prognostic values [9–11] as an inability to conceive despite normal semen parameters is common [12].

Therefore, a wealth of research has focused on improving the efficiency of sperm selection techniques, as well as the development of functional assays designed to select sperm based on preferred characteristics to optimise treatment outcomes. An example of this is sperm-hyaluronic acid binding (HAB).

Hyaluronic acid (HA) is abundant in vivo and acts as a physiological selector of mature sperm with low levels of DNA fragmentation, as described elsewhere [13–15]. Based on this selective ability, a diagnostic tool has been developed—the sperm hyaluronan-binding assay (HBA) [16]—where the number of HA-bound and unbound sperm are calculated in each sample. It is reported that the proportion of sperm with fertilising ability is predictive of the likelihood of successful outcomes in conventional IVF [17,18] and ICSI cycles [19]. It is suggested that much like the Hemi-Zona Assay, HBA may be used as a diagnostic supplement to traditional semen analysis to assist with treatment modality selection, while eliminating the need for assays requiring the human zona pellucida (ZP).

HA-binding may also be used to select mature sperm for insemination (Figure 1). HA-rich viscous media, such as SpermSlow (CooperSurgical, Måløv, Denmark), represent alternatives to polyvinylpyrrolidone (PVP) during the sperm-immobilisation stages of ICSI. As HA-containing products are not associated with adverse effects on post-injection embryo development and can be metabolised by the oocyte [20–22], it is unlikely that introduction to the ooplasm during injection would compromise embryonic viability. Studies have reported a similar effectiveness of HA media when compared to PVP, without the associated defects in embryo development [23,24]. Moreover, there are reports that HA media may reduce the incidence of early pregnancy loss in ICSI cases [25–27].

Commercially available physiological ICSI (PICSI) dishes, containing micro-drops of dense HA media, may also be used in sperm selection prior to ICSI [28,29]. A number of studies report varying increases in fertilisation, embryo development, implantation, and clinical pregnancy rates following ICSI insemination with HA selected sperm. The results are, however, inconsistent between studies [28,30]. Large multicentre randomised controlled trials (RCTs) reported no statistically significant difference in implantation or clinical pregnancy rates when compared with conventional ICSI, however significant decreases in the incidence of pregnancy loss were identified [31,32].

Recent Cochrane Database reviews reported that the evidence from RCTs thus far was insufficient to draw any clear conclusions on the efficacy of HA-binding to improve clinical pregnancy or live birth rates [33–35]. However, these reviews, along with other large-scale studies, did report significant decreases in miscarriage rates following insemination with HA selected sperm compared with conventional ICSI [26,31,33–35]. Moreover, no studies report adverse effects on clinical outcomes following HA-sperm selection. Notwithstanding, HA-binding not being recommended for routine clinical use is a matter of heated debate [31,36,37].

The Human Fertilisation and Embryology Authority (HFEA), the UK government regulator of ART and embryological research, published a consensus statement on the use of IVF "add-on" treatments [38], including PICSI. This was an attempt to guide both professionals and patients on how the use of such "add-ons" should be approached to protect patients from financial exploitation. A traffic light system was launched to present their evaluation of the evidence regarding the safety and effectiveness of a number of commonly utilised "add-on" laboratory and clinical treatments, typically offered at additional cost.

A red light, as assigned to PICSI, signifies that no evidence of safety or effectiveness, as reported by an RCT, supports the routine use of the treatment [39]. While this is not a ban on the use of these adjuvant treatments, allowing some patient autonomy, it reflects the HFEA's stance that this treatment should not be offered to patients at additional cost under the guise of increasing the likelihood of live birth, and should strictly be applied only in a research setting. However, due to the lack of green lighting in the scheme, it is difficult to envisage how clinical guidelines on which specific infertility aetiologies may benefit from *any* add-on treatment might proceed. This is much needed to aid clinicians in

their recommendations of any treatment supplement to specific patient groups, e.g., those suffering recurrent fertilisation failure or recurrent pregnancy loss.

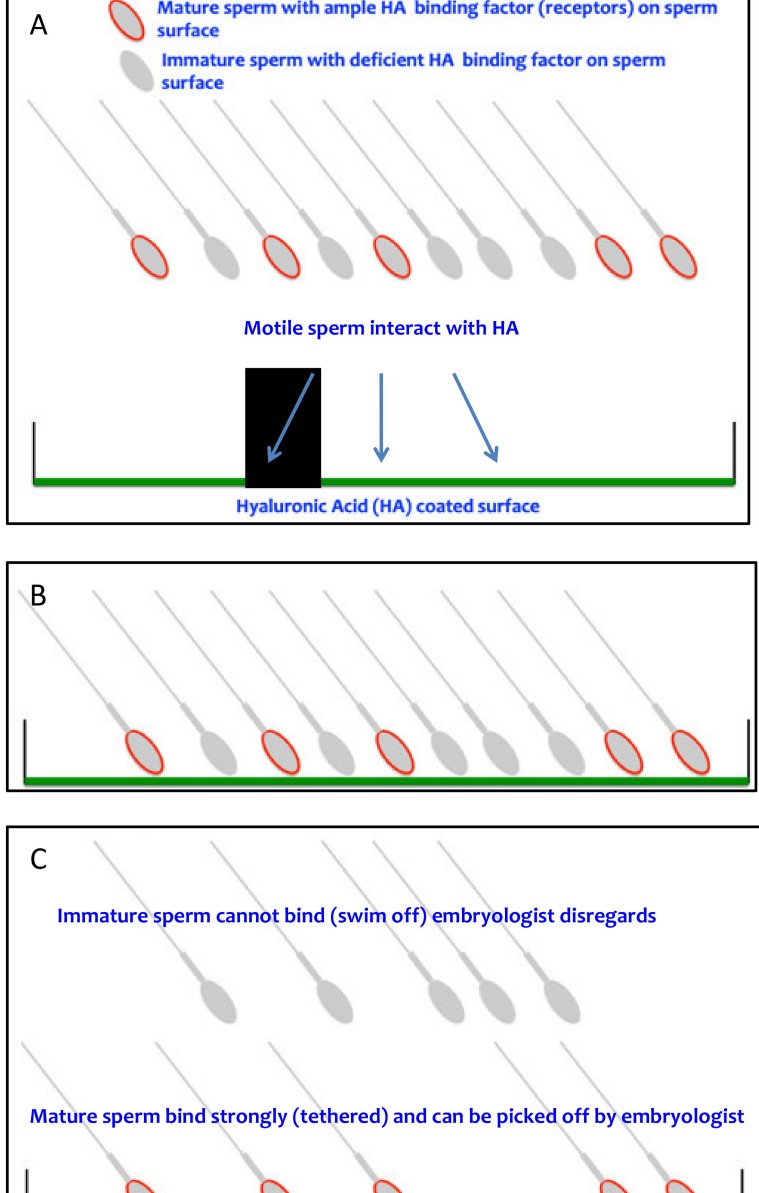

**Figure 1.** The clinical use of hyaluronic acid binding (HAB)-sperm selection, depicting the introduction of sperm to HA-coated dishes (**A**,**B**) and the binding of mature sperm to HA (**C**). The bound sperm are selected for intracytoplasmic sperm injection (ICSI).

Despite evidence of the potential benefits of HAB, there remains no consensus on which patient groups, if any, would benefit the most, even in a research setting. Additionally, there are no known practice guidelines on how such patients should be managed by the clinical team. This study seeks to determine whether the evidence, as it stands, is sufficient to dismiss the routine use of HAB in the clinical setting or accept that adoption of such techniques may benefit specific sub-populations of patients. Specifically, the questions posed to frame this issue are: How robust is the evidence that HAB-sperm selection:

- Provides prognostic information regarding the likelihood of success of treatment via HAB scores? (Study question 1)

- Improves the incidence of miscarriage/pregnancy loss following insemination with HAB selected sperm compared with conventional insemination techniques? (Study question 2)
- Improves clinical outcomes (e.g., live birth rates) compared with conventional IVF/ICSI in the unexplained infertility population? (Study question 3)

## 2. Materials and Methods

A systematic search of published studies up to November 2020 was performed following Preferred Reporting Items for Systematic Reviews and Meta-Analysis (PRISMA) guidelines [40]. PubMed, Medline, and the Cochrane Databases were searched. The Embase database was not included as access was inconsistent.

Other searches included:

- Cochrane Central Register of Controlled Trials (CENTRAL)
- US National Institutes of Health Ongoing Trials Register (ClinicalTrials.gov)
- Hand-searching through reference lists and citing articles of selected studies. This review was not registered.

### 2.1. Search Terms for Each of the Research Questions

The following research questions were included:

- Is sperm HAB score prior to insemination predictive of clinical outcomes?
- Does sperm selection by HAB reduce the rate of pregnancy loss in IVF or ICSI cycles?
- Does sperm selection by HAB improve clinical outcomes for all infertility patients?

The following keywords, as well as respective combinations, were chosen: ("hyaluronic acid binding" or "hyaluronan binding"), ("sperm" or "spermatozoa"), ('IVF' or "in vitro fertilization" or 'ICSI' or "intracytoplasmic sperm injection"), ("predict" or "predictive" or "prognostic"), ("miscarriage" or "pregnancy loss" or "abortion"), ("PICSI" or "physiological ICSI").

### 2.2. Types of Studies

The searches were restricted to English language and human only studies, including clinical trials, RCTs, meta-analyses, and systematic reviews. Non-randomised, retrospective, and observational studies were also included to ensure all evidence on the topic was reviewed.

The titles and abstracts of all identified studies were assessed, and once deemed relevant, the full-text articles were retrieved. The reviewers then assessed the full text while searching the reference lists for additional studies. Studies were excluded if they were not published as full-text articles, e.g., abstract only articles and posters. The main study outcomes of interest were fertilisation rate, clinical pregnancy rate (defined as the presence of a gestational sac on ultrasound), pregnancy loss rate (PLR, defined as miscarriage per clinical pregnancy), and live birth rate. Studies which did not report either fertilisation rate (FR), clinical pregnancy rate (CPR), pregnancy loss rate (PLR), or live birth rate (LBR) were not included. Other outcomes of interest included cleavage rate, embryo quality, and implantation rate. Studies which reported at least one of the following for the outcomes of interest were included: odds ratios (OR), relative risk (RR), or mean difference.

### 2.3. Types of Participants

The following exclusion criteria were also applied:

- Advanced maternal age (>43 years)
- Use of donated gametes
- Use of surgically retrieved sperm

### 2.4. Quality and Risk of Bias Assessments

The quality of the study methodology was assessed using a modified "Downs and Black standardised checklist", rating items across study quality, external validity, bias, and

confounding factors and selection bias [41]. This checklist was combined with a more recent checklist by Meader et al. [42], which is designed to enhance the reproducibility of GRADE assessments, the format utilised by Cochrane Reviews [43]. This was then used to present a comprehensive "Traffic Light" diagram to report the relative quality/risk of bias for each study.

## 3. Results

### 3.1. Can Assessment of Sperm Function by HA-Binding Predict Clinical Outcomes?

The initial database search yielded 71 results, while cross-referencing of relevant articles identified 3 additional articles. Then, 28 articles were deemed relevant and the full texts were retrieved for assessment, leading to the exclusion of 20 articles (see Figure 2), and the inclusion of 8 studies. The inclusion of studies without any randomisation of subjects was deemed appropriate considering that the HAB score procedure is diagnostic and will not introduce any bias. Included studies are summarised in Table 1. A summary of quality and bias assessment results is provided in Figure 3.

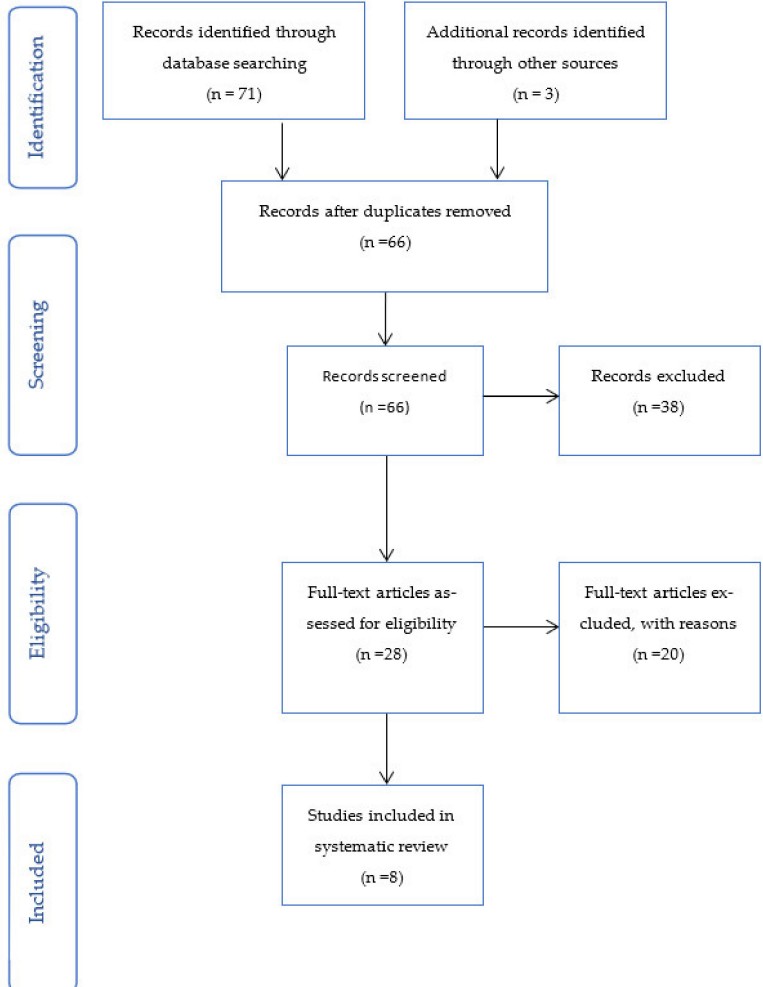

**Figure 2.** A PRISMA diagram demonstrating the study selection process for study question 1. This flowchart outlines the search and selection process employed in study question 1—Is sperm HAB score prior to insemination predictive of clinical outcomes?

**Table 1.** A table summarising the studies included in this review for the assessment of the predictive value of sperm-HA binding: study question 1—Is sperm HAB score prior to insemination predictive of clinical outcomes? * Indicates a significant result ($p < 0.05$). FR: fertilisation rate; CPR: clinical pregnancy rate; LBR: live birth rate; PLR: pregnancy loss rate; IR: implantation rate; BPR: biochemical pregnancy rate; IVF: in vitro fertilisation.

| Author | Intervention | Study Type | Indication | Reported Outcomes | Results |
|---|---|---|---|---|---|
| *Esterhuizen, Franken et al. (2015)* [44] | ICSI | Prospective, Controlled | Mild–Moderate endometriosis | HAB score and FR, BPR, and CPR | R = 0.60 ($p$ = 0.0001 *), R = 0.24 ($p$ = 0.02 *), R = 0.14 ($p$ = 0.14) |
| *Kovacs, Kovats et al. (2011)* [45] | IVF/ICSI | Prospective, Blinded Randomised Controlled | Unexplained infertility Female age <40 Normozoospermic patients | HAB score and FR | No significant correlation (data not presented) |
| *Miller et al. (2019)* [31] | PICSI | Prospective, blinded, randomised, controlled | Unexplained infertility Female age 18–38 Patients able to produce fresh ejaculate on the day of OR | HAB score, FR, CPR, PLR, LBR | No significant correlations reported (data not presented) |
| *Mokánszki, Tóthné et al. (2014)* [46] | PICSI | Prospective Controlled Non-randomised (treatment allocation based on HAB score) | Female age <40 years Sperm concentration >10,000/mL on the day of OR | FR | 64.5% vs. 56.5% ($p$ > 0.05) HAB score >60%:73.36% vs. 60.1% ($p$ < 0.05 *) HAB score ≤60%:55.7% vs. 52.8% ($p$ > 0.05) |
| | | | | IR | 21.7% vs. 17.12% ($p$ > 0.05) HAB score >60%:20.8% vs. 21.47% ($p$ > 0.05) HAB score ≤60%:22.6% vs. 12.76% ($p$ < 0.05 *) |
| | | | | CPR | 40.46% vs. 29.22% ($p$ < 0.05 *) HAB score >60%:41.67% vs. 31.85% ($p$ < 0.05 *) HAB score ≤60%:39.3% vs. 26.6% ($p$ < 0.05 *) |
| | | | | PLR | 2% vs. 5.14% ($p$ < 0.05 *) HAB score >60%:2.2% vs. 8.37% ($p$ < 0.05 *) HAB score ≤60%:1.99% vs. 1.9% ($p$ > 0.05) |
| | | | | LBR | 0.45% vs. 0.42% ($p$ > 0.05) HAB score >60%:0.42% vs. 0.58% ($p$ > 0.05) HAB score ≤60%:0.49% vs. 0.27% ($p$ < 0.05 *) |
| *Pregl Breznik, Kovačič et al. (2013)* [47] | IVF/ICSI | Prospective | Mild male factor Unexplained infertility Female factor infertility | HAB score and FR | FR <50%, HAB 85.1% vs. FR >50%, HAB 93% ($p$ = 0.019 *) |
| *Said and Land (2011)* [48] | | Systematic Review | | HAB score, FR, CPR, | No significant correlations reported |
| *Worrilow, Eid et al. (2013)* [26] | PICSI | Prospective, Double blinded, randomised, controlled | Female age <40 HAB score >2% Sperm concentration >10,000/mL | IR | 33.5% vs. 32.2% ($p$ > 0.05) HAB score >65%:37.9% vs. 34.8% ($p$ > 0.05) HAB score ≤65%:37.4% vs. 30.7% ($p$ > 0.05) 47.3% vs. 47.8% ($p$ > 0.05) |
| | | | | CPR | HAB score >65%:46.2% vs. 51.1% ($p$ > 0.05) HAB score ≤65%:50.8% vs. 37.9% ($p$ > 0.05) |
| | | | | PLR (HAB score unwashed) | 4.3% vs. 10% ($p$ > 0.05) |
| | | | | PLR (HAB score washed) | HAB score >65%:5.3% vs. 3.5% ($p$ > 0.05) HAB score ≤65%:3.3% vs. 15.1% ($p$ = 0.021 *) 4.3% vs. 10% ($p$ > 0.05) HAB score ≤65%:0% vs. 18.5% ($p$ = 0.016 *) |

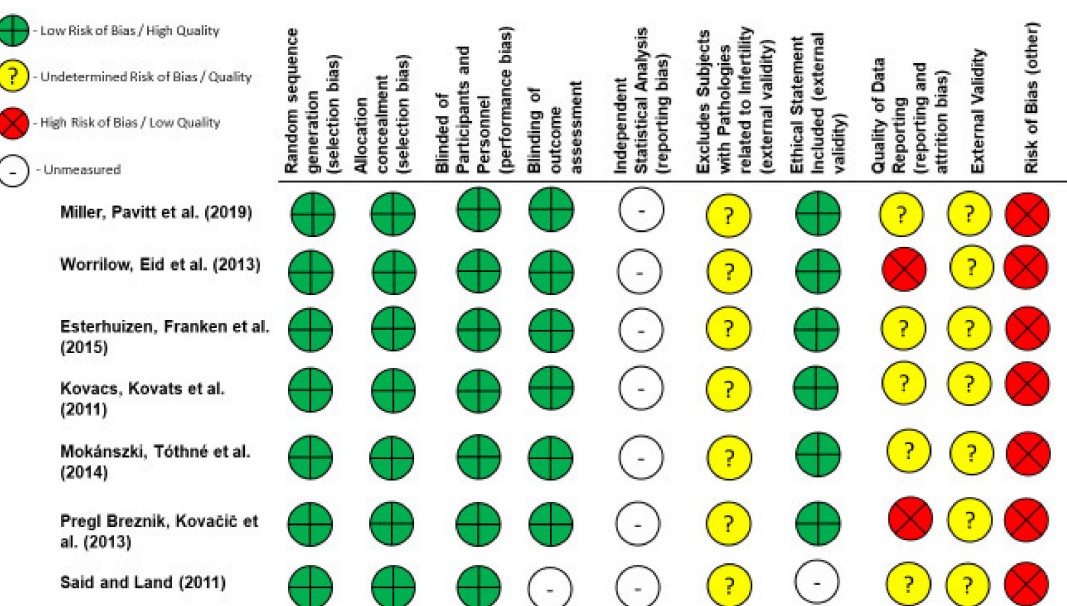

**Figure 3.** A summary of the authors' judgement on quality and bias assessment results of the 7 reviewed studies for study question 1—Is sperm HAB score prior to insemination predictive of clinical outcomes? Green: low risk of bias/high quality, Yellow: undetermined risk of bias/quality, Red: high risk of bias/low quality, White: assessment does not apply. Studies were considered at low risk of biases related to selection and performance given that no intervention was involved in the study groups, and procedures were of a diagnostic nature without interfering with patient treatment. All studies were considered of undetermined quality given that exclusion criteria were either ill-defined or included pathologies which may impact later outcomes, e.g., endometriosis. [26,31,44–48].

### 3.1.1. Fertilisation Rate (FR)

Six of the included studies reported fertilisation rates [31,44–48]. Esterhuizen et al. calculated the HAB score of raw semen samples in 91 couples undergoing conventional ICSI, reporting a significant positive correlation between HAB score and fertilisation rates ($R = 0.60$, $p = 0.0001$) [44]. Breznik et al. reported similar results in a prospective, controlled study of 133 couples diagnosed with male factor or unexplained infertility, where HA-binding assessments were carried out on density gradient centrifugation (DGC)-prepared sperm used in IVF cycles only [47]. Moreover, when results were analysed using ROC curves regarding achieving 50% fertilisation rates, the curves were statistically significant ($p = 0.007$), and fertilisation > 50% was associated with significantly higher HAB scores when compared with <50% ($p = 0.019$). However, no statistically significant associations were reported in ICSI cycles.

A similar trend was reported by Mokánszki et al., though this did not reach statistical significance ($R = 0.53$, $p \geq 0.05$) [46]. While fertilisation rates were compared between the study (PICSI) and control (ICSI) groups and significant differences were identified, this analysis was not extended to compare rates between the different categories of HAB score analysed in this study (>60%, <60%, >70%, <50%) across both treatment groups. Therefore, while rates differ between the categories, it is not determined whether these differences are significant. While Miller et al. reported fertilisation rates, no significant correlation between HAB score and FR was reported [31]. The remaining studies reported no statistically significant correlations.

### 3.1.2. Clinical Pregnancy Rates (CPR)

Seven of the included studies reported clinical pregnancy rates [19,26,31,44–46,48]. Three studies reported no statistically significant correlations between HAB score and clinical pregnancy rates [44,45,48]. While one study reported a significant correlation

between HAB scores and CPRs [19], they were unable to verify the validity of the reported 65% lower binding limit. Three studies reported CPRs with patients stratified by HAB score, however analysis was not carried out to determine the effects of HAB score on CPRs [26,31,46]. Therefore, no significant relationship between HAB scores and clinical pregnancy rates was identified.

### 3.1.3. Live Birth Rates (LBR)

One study, a prospective, randomised, double-blind controlled trial assessing 2752 couples undergoing conventional ICSI or PICSI, reported live birth rates [31]. However, they reported no differential effect of HAB score on the incidence of live birth between the two treatment groups.

### *3.2. Is the Incidence of Miscarriage/Pregnancy Loss Reduced following Insemination with HAB Selected Sperm Compared with Conventional Insemination Techniques?*

The initial database search yielded 82 results, while cross-referencing of relevant articles identified 8 additional articles. Then, 23 articles were deemed relevant, and the full text was retrieved for assessment, leading to the exclusion of 17 articles (see Figure 4), and the inclusion of 5 studies. Included studies are summarised in Table 2. This included four prospective randomised controlled trials [26,28,31,49], two of which were blinded [26,31], and one Cochrane systematic review [34]. A summary of the results of quality and bias assessments is provided in Figure 5.

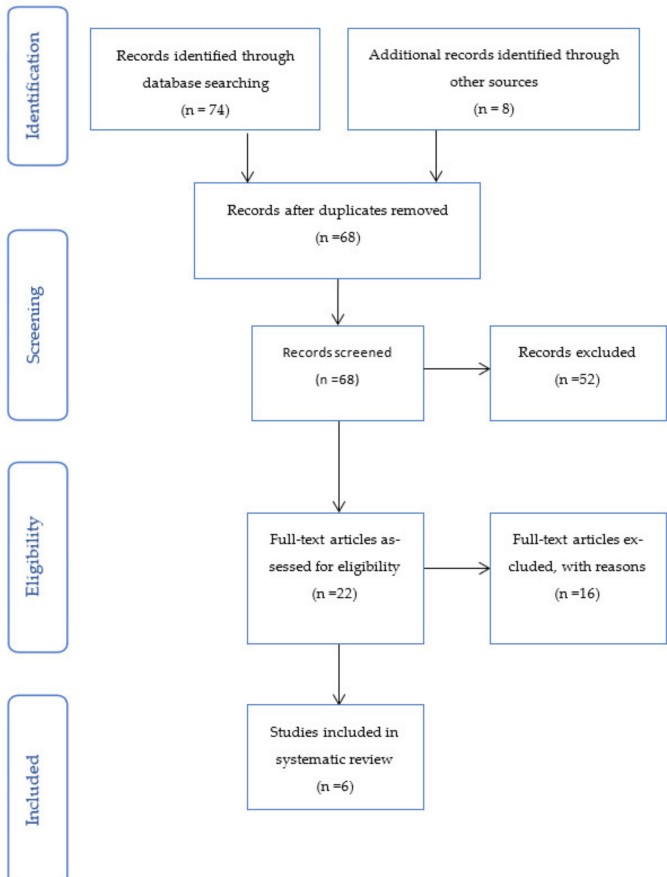

**Figure 4.** A PRISMA diagram demonstrating the study selection process for study question 2—Does sperm selection by HAB improve the rate of pregnancy loss in IVF or ICSI cycles?

**Table 2.** A table summarising the studies included in this review for study question 2—Does sperm selection by HAB improve the rate of pregnancy loss in IVF or ICSI cycles? * Indicates a significant result ($p < 0.05$). MACS—magnetic activated cell sorting. OR: Odds Ratio; RR: Risk Ratios.

| Author | Intervention | Control | Study Type | Indication | Reported Outcomes | Results |
|---|---|---|---|---|---|---|
| *Lepine, McDowell et al. (2019)* [34] | PICSI | ICSI | Systematic Review | Unexplained Infertility | Miscarriage per woman randomly assigned | 43 of 1000 vs. 70 of 1000 (RR = 0.61) |
| | | | | | Miscarriage per Clinical Pregnancy | 122 per 1000 vs. 197 per 1000 (RR = 0.62) |
| *Miller et al. (2019)* [31] | PICSI | ICSI | Prospective, blinded, randomised, controlled | Unexplained infertility Female age 18-38 Patients able to produce fresh ejaculate on the day of OR | Miscarriage per Clinical Pregnancy | 4.3% vs. 7% (OR = 0.61, $p = 0.003$) |
| *Parmegiani, Cognigni et al. (2010)* [28] | SpermSlow | ICSI | Prospective, randomised, controlled | No female age range reported Motility >5%, Sperm concentration >1 million/mL | Miscarriage per Clinical Pregnancy | 18.2% vs. 19.3% ($p > 0.05$) |
| *Troya and Zorrilla (2015)* [49] | PICSI | MACS and ICSI | Prospective, Randomised, Controlled | Unexplained infertility Normozoospermic Female age >35 | Miscarriage per Clinical Pregnancy | 5.3% vs. 5.5% and 13.3% ($p > 0.05$) |
| *Worrilow, Eid et al. (2013)* [26] | PICSI | ICSI | Prospective, Double blinded, randomised, controlled | Female age <40 HAB score >2% Sperm concentration >10,000/mL | Miscarriage per Clinical Pregnancy | 4.3% vs. 10 % ($p > 0.05$) |
| | | | | | Miscarriage per Clinical Pregnancy, Final HAB score >65% | 5.9% vs. 7.8% ($p > 0.05$) |
| | | | | | Miscarriage per Clinical Pregnancy, Final HAB score ≤65% | 0% vs. 18.5% ($p = 0.016$ *) |
| *Majumdar and Majumdar (2013)* [50] | PICSI | ICSI | Prospective, randomised, controlled | Unexplained infertility, female patients <43, >3 oocytes collected | Miscarriage per clinical pregnancy | 12% vs. 25% ($p > 0.05$) |

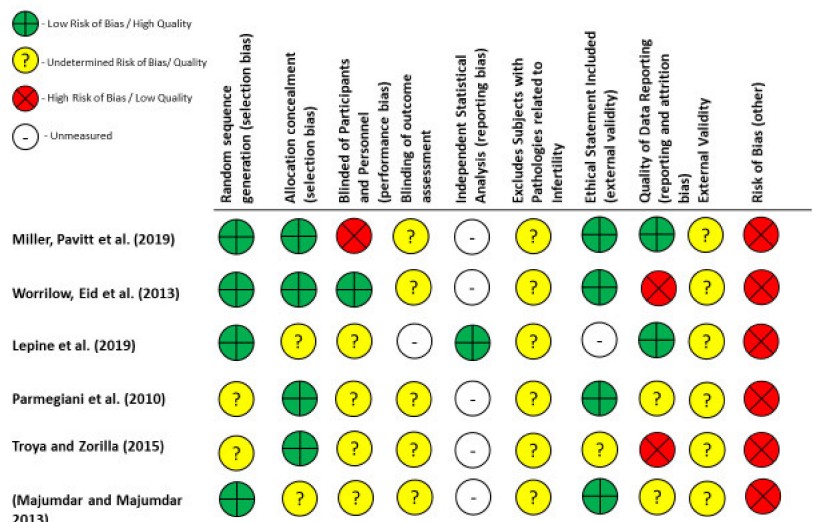

**Figure 5.** A summary of the authors' judgement on quality and bias assessment results on the 5 studies included in study question 2—Does sperm selection by HAB reduce the rate of pregnancy loss in IVF or ICSI cycles? Green: low risk of bias/high quality, Yellow: undetermined risk of bias/quality, Red: high risk of bias/low quality, White: assessment does not apply. [26,28,31,34,49,50].

Three studies reported small decreases in PLR when comparing HA-ICSI, using either HA-rich media [28], or PICSI [49,50] with conventional ICSI, however these failed to reach statistical significance. However, a large-scale, multicentre trial including 2752 couples reported a significant decrease of 2.7% in PLRs in PICSI cycles compared to ICSI (OR = 0.61, *p* = 0.0003) [31]. A second, smaller study [26], which employed the use of HA-rich viscous media for sperm selection, reported similar trends, however this only reached statistical significance when data were stratified by HAB score (<65%) (see Table 2). This difference remained significant when HAB scores from both raw and prepared semen were analysed.

A Cochrane systematic review of advanced sperm selection techniques in an unselected infertility population also reported pregnancy loss [34]. They reported that, based on low-quality evidence from three RCTS, including 3005 cycles, miscarriage per woman randomly assigned was reduced from 70 to 43 per 1000 participants (RR = 0.62) when comparing ICSI with HA-ICSI. Moreover, they also reported miscarriage per clinical pregnancy based on low-quality evidence from 3 RCTs, including 1065 cycles, and estimated a reduction from 197 to 122 per 1000 participants (RR = 0.62).

*3.3. Can HAB-Sperm Selection Improve Clinical Outcomes Compared with Conventional IVF/ICSI in the Unexplained Infertility Population?*

The initial database search yielded 47 results, while cross-referencing of relevant articles identified 33 additional articles. Then, 28 articles were deemed relevant, and the full texts were retrieved for assessment, leading to the exclusion of 20 articles (see Figure 6), and the inclusion of 8 studies, including 2 systematic reviews and 6 prospective randomised studies. Included studies are summarised in Table 3. A summary of quality and bias assessment results is provided in Figure 7.

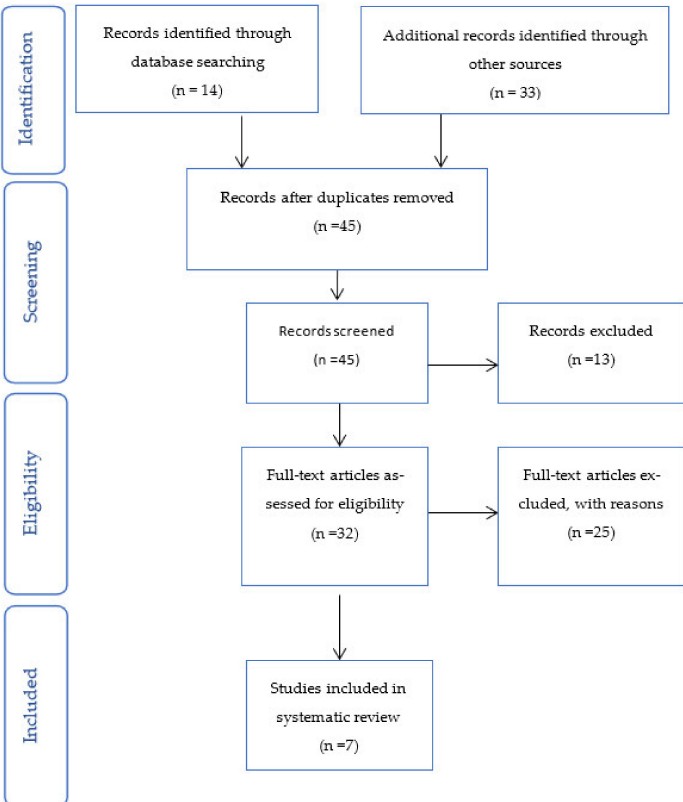

**Figure 6.** A PRISMA diagram summarising the literature search and selection process for study question 3—Does sperm selection by HAB improve clinical outcomes for all infertility patients?

**Table 3.** A table summarising the studies included in this review for study question 3, involving the assessment of the effect of HAB-sperm selection on clinical outcomes in the unexplained fertility population. EQR: rate of top-quality embryo formation; EDR: embryo development rate.

| Author | Intervention | Control | Study Type | Indication | Reported Outcomes | Results |
|---|---|---|---|---|---|---|
| *Beck-Fruchter, Shalev et al. (2016)* [51] | HA-ICSI | | Systematic Review | Unexplained Infertility | FR, CPR | No significant difference |
| | | | | | Cleavage rate | RR = 0.94 in favour of Control ($p = 0.0001$) |
| | | | | | EQR | 35–36% vs. 22–24% ($p < 0.05$) RR = 1.53 in favour of HA-ICSI ($p < 0.0001$) |
| | | | | | IR | RR 1.34 in favour of HA-ICSI ($p = 0.24$) |
| | | | | | PLR | No significant difference (data not presented) |
| | | | | | LBR | No significant difference (data not presented) |
| *Miller et al. (2019)* [31] | PICSI | | Prospective, blinded, randomised, controlled | Unexplained infertility Female age 18–38 Patients able to produce fresh ejaculate on the day of OR | FR, CPR, LBR PLR | No significant difference (data not presented) 4.3% vs. 7.0% ($p = 0.003$) |
| *Worrilow, Eid et al. (2013)* [26] | PICSI | | Prospective, Double blinded, randomised, controlled | Female age <40 HABScore >2% Sperm concentration >10,000/mL on the day of OR | IR | 33.5% vs. 32.2% ($p > 0.05$) |
| | | | | | CPR | 47.3% vs. 47.8% |
| | | | | | PLR | 4.3% vs. 10% ($p > 0.05$) |
| *Lepine, McDowell et al. (2019)* [34] | PICSI | | Systematic Review | | LBR per patient | RR = 1.09 in favour of HA-ICSI ($p > 0.05$) RR = 0.62 in favour of HA-ICSI ($p > 0.05$) |
| | | | | | PLR per Clinical Pregnancy CPR per patient | No significant difference (data not presented) |
| *Parmegiani, Cognigni et al. (2010)* [28] | SpermSlow | | Prospective, randomised, controlled | No female age range reported Motility >5%, Sperm concentration >1 million/mL on the day of OR | FR, CPR, PLR EDR | No significant differences (data not presented) 95.0% vs. 84.0% ($p = 0.001$) |
| *Troya and Zorrilla (2015)* [49] | PICSI | ICSI and MACS | Prospective, randomised, controlled | Endometriosis Excluded Normozoospermic patients undergoing ICSI | FR | 70.15% vs. 78.97% and 80.28% ($p = 0.036$) |
| | | | | | CLR | 40.4% vs. 27.3% and 58.1% ($p = 0.019$) |
| | | | | | PLR per Clinical Pregnancy | 5.3% vs. 13.3% and 5.5% ($p > 0.05$) |
| *Majumdar and Majumdar (2013)* [50] | PICSI | ICSI | Prospective, randomised, controlled | Female patient <43 years Unexplained infertility | Pregnancy FR, CPR, PLR, LBR | No significant difference reported |

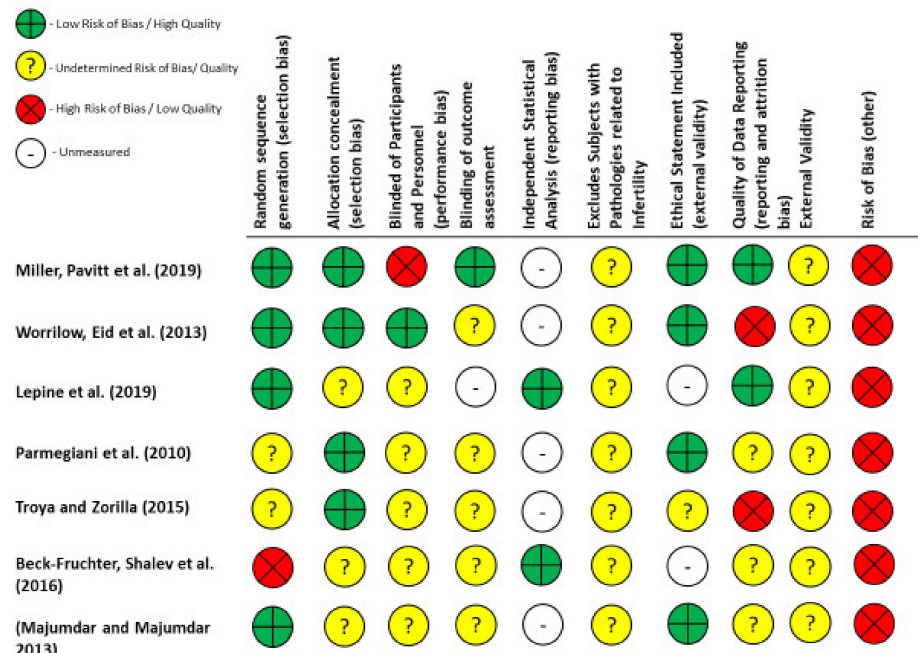

**Figure 7.** A summary of the authors' judgement on quality and bias assessment results for studies included in study question 3—Does sperm selection by HAB improve clinical outcomes for all infertility patients? Green: low risk of bias/high quality, Yellow: undetermined risk of bias/quality, Red: high risk of bias/low quality, White: assessment does not apply. [26,28,31,34,49–51].

### 3.3.1. Fertilisation Rates (FR)

Five studies reported FR, including one systematic review [51] and four RCTs [28,31,49,50]. Beck-Fruchter et al. reported that despite analysing the results of 9700 injected oocytes from 7 studies, there was no association between HA sperm selection and FRs (RR 1.02, 95% CI 0.99 to 1.06) [51]. While FRs were higher in HA-ICSI in 1 RCT assessing 232 couples [28], this did not reach statistical significance. Similarly, Miller et al. and Majumdar et al. did not report any statistical differences in FRs between the treatment groups [31,50].

Conversely, 1 study assessing the effects of both magnetic activated cell sorting (MACS) (see Table 3) and PICSI compared with conventional ICSI in 135 couples undergoing ART reported a decrease in FRs in the PICSI group compared with the ICSI and MACS treatment groups (70.15% vs. 78.97% and 80.28%, respectively, $p = 0.036$) [49]. Therefore, there is no clear correlation between FRs and HA-ICSI.

### 3.3.2. Clinical Pregnancy Rates (CPR)

Seven studies reported CPRs, including two systematic reviews [34,51] and five RCTs [26,28,31,49,50]. Beck-Fruchter et al. reported no difference in CPRs per cycle when comparing HA-ICSI and conventional ICSI (RR 1.10, 95% CI 0.93 to 1.29) [51]. Similarly, based on low-quality evidence in 4 RCTs including 3492 patients, Lepine et al. found no significant differences in CPRs between treatment groups (RR 1.00, 95% CI 0.92 to 1.09) [34]. Four RCTs reporting CPRs also failed to report statistically significant differences [26,28,31,50]. While Troya et al. reported a significant difference in CPRs when comparing MACS, ICSI, and PICSI (58.1% vs. 27.3% and 40.4%, $p = 0.019$), only the difference between MACS and ICSI groups was noted as significant [49]. Therefore, no clear correlation between CPRs and HA-ICSI is reported.

### 3.3.3. Live Birth Rates (LBR)

Three studies reported LBRs, including one systematic review [34] and two RCTs [31,50]. Miller et al., reported an increase in LBR when comparing PICSI and ICSI in 2752 patients,

however this did not reach statistical significance (27.4% vs. 25.1% per woman randomised, RR = 1.12, *p* = 0.18) [31]. Lepine et al. [34], based on low-quality evidence from two RCTs including the abovementioned papers [31,50], came to a similar conclusion, reporting that little or no difference is observed between study groups per woman randomised (RR = 1.09, 95% CI 0.97 to 1.23, 2903 patients). Therefore, it remains unclear whether a correlation between LBRs and HA-ICSI exists.

## 4. Discussion

A main objective of this study was to determine the prognostic value of HAB scores prior to insemination, however, considering the variation in assessment methods, e.g., assessing raw vs. prepared samples, this remains unclear. While the trends were towards improved clinical pregnancy live birth rates following HAB-sperm selection, these differences were not statistically significant. However, this study was able to fulfil the third objective in identifying a significant difference in the incidence of miscarriage following insemination using HAB-sperm.

### 4.1. Confounding Variables

Inconsistency in sperm-HA binding assessment methodology was identified as a confounder in reviewed studies, and may account for the conflicting evidence presented, particularly as most studies carried out HAB assessments on sperm prepared by various methods [31]. While studies have reported that preparation may improve the concentration of sperm with HA-binding ability [19,52,53], the shear forces involved in multiple centrifugation steps may lead to reduced membrane stability, increased DNA fragmentation, and the generation of reactive oxygen species (ROS) [54–61]. Such factors are reported to have a detrimental impact to early treatment outcomes, such as fertilisation and implantation rates [23,24,62–64], and may account for conflicting reports of effects on CPRs and PLRs, as this study has identified no association between HAB score and these outcomes, while many studies failed to report a detailed sub-group analysis of the sperm preparation methods used [26,31,46].

An additional confounder identified in this study was a variation in embryo culture and transfer protocols: all studies carried out both Day 3 and Day 5 embryo transfers, contingent on embryo availability and morphology. While cleavage stage embryo transfer on Day 3 is effective [65], blastocyst transfer on Day 5 may be more representative of in vivo fertilisation and implantation as embryos typically reach the uterus around Day 4 post-fertilisation [66], and is also reported to significantly improve implantation, clinical pregnancy, and live birth rates, when compared with cleavage stage transfer [67]. However, the frequency of Day 3 and 5 embryo transfers was not reported in numerous studies, nor were the groups compared for differential effects of embryo transfer protocols.

A recurring issue during the review of the literature was the inclusion of, or failure to exclude, female factor infertility. Mild–moderate endometriosis is associated with poor oocyte and embryo development, while severe endometriosis is associated with significantly lower implantation and clinical pregnancy rates [68]. The mechanisms by which endometriosis impacts fertility are complex and dynamic, including altered inflammatory responses and local action of inflammatory cytokines [68]. Moreover, tubal factors such as hydrosalpinxes [69,70] or uterine fibroids [71] are also associated with poorer clinical outcomes. Therefore, the failure to exclude these conditions may reduce the validity of the reported results.

### 4.2. HAB Score and Clinical Outcomes

Significant associations were identified between high fertilisation rates (>50%) and HAB scores [44,47]. While this was observed in IVF cycles only in one study [47], similar associations may not arise in ICSI cycles as, despite a higher frequency of sperm with HA-binding ability in high HAB score samples, sperm selection remains dependent on the morphological assessment of the embryologist. Therefore, sperm of reduced binding

ability and fertilising capacity may still be selected for ICSI insemination. However, this is important to note as the incorporation of sperm selection methods aimed at increasing the yield of sperm with a high binding capacity into daily practice would reduce the likelihood of this occurring. Moreover, this study calculated HAB scores of sperm prepared by DGC [47]. A similar study [44] reported a significant association between HAB score and fertilisation rates in ICSI cycles, however these results were based on the HA-binding ability of sperm from unprepared semen samples. This suggests that preparation of sperm by DGC may impact the binding ability of sperm.

Based on these results, the specificity of the HAB score as a predictive test of fertilisation may be reduced in ICSI cycles due to the embryologists' autonomy over sperm selection, as it is generally accepted that ICSI increases the likelihood of injecting oocytes with functionally inferior sperm, e.g., defective in phospholipase C zeta (PLC ζ) content, essential for oocyte activation and successful fertilisation [72] or of low genomic integrity [73]. However, it may present a cheap, effective diagnostic tool in predicting the fertilising ability of sperm in IVF cycles, as it provides a quantification of the proportion of sperm likely to penetrate the cumulus oocyte complex (COC) and ZP, steps essential for fertilisation both in vivo and IVF. Therefore, further studies focusing on IVF cycles only may not only provide further evidence of its applicability to treatment modality selection, but also aid in excluding defective sperm-binding factors in cases of total fertilisation failure, providing information vital to the success of future treatment.

### 4.3. HAB-Sperm Selection and Pregnancy Loss

While two studies reported no significant difference in PLRs between HA-ICSI and conventional ICSI [28,49], there were significant limitations to these studies. Troya et al. included only normozoospermic patients, excluding male factor infertility, which represents the patient group arguably set to benefit most from the intervention [74], and failing to report sperm morphology parameters of both groups. Parmegiani et al. not only failed to describe exclusion of specific female factor infertility, but the study was also limited to three oocytes injected per patient, as well as the transfer of all produced embryos regardless of morphology or quality, in accordance with Italian law [28]. This reduces the validity and generalisability of these results.

Conversely, a significant reduction in PLRs was identified from three of the included studies [26,31,34]. While this only reached statistical significance in one study following stratification of the patients according to HAB score (>/<65%) [26], a meta-analysis by a Cochrane review identified a significant decrease in PLR across patient groups [34]. While this reduction was not reflected in the LBR, the reported reduction from 70 to 43 per 1000 women randomly assigned remains relevant, as this may not only lead to a reduction in the emotional distress induced by miscarriage but could also increase the cumulative live birth rate following serial embryo transfers in further studies. Further analysis of Miller et al. has also revealed that the reduction in miscarriage was more pronounced in female patients over 35 years of age [75]. It is suggested that as oocyte quality diminishes with maternal age, so too does the competence of oocyte-derived DNA repair mechanisms. Therefore, a reduction in the burden of DNA damaged spermatozoa could have contributed to the reduction in miscarriage rates. Given that this patient group represents a growing demographic of patients seeking ART, these results are profoundly important in steering future practice.

### 4.4. HAB-Sperm Selection and Clinical Outcomes

The literature on fertilisation and HAB-sperm selection is inconsistent. It is reported that sperm with the capacity bind to HA possess superior chromatin structure and high DNA integrity, which is predictive of fertilising potential. While one study reported a significant reduction in PICSI fertilisation rates compared with MACS, this was not the case when compared with conventional ICSI [49]. Another report of reduced fertilisation rates following PICSI emerged from mechanistic analysis of the HABSelect trial results [75],

however this was not subsequently reflected in a reduction in clinical pregnancy or live birth rates [76]. Moreover, a 2019 Cochrane review reported no statistically significant difference in fertilisation rates between HAB-sperm selection and conventional methods [34]. The results of the current review are therefore in line with the literature as it currently stands, that there is no significant difference in fertilisation rates when HAB-sperm selection is employed.

One study reported higher CPRs in the intervention groups (PICSI and MACS) than the conventional ICSI group, however this did not reach statistical significance in the case of PICSI [49]. This not only highlights that patients with male factor infertility benefit from an intervention or advanced sperm selection technique, but also that there are no detrimental effects of the use of HAB-sperm selection when compared with routine ICSI practice. The remaining studies reported trends towards improved CPRs and LBRs, however none reached statistical significance. Therefore, the results of this study are in agreement with the current literature, that there is no significant difference in CPRs or LBRs when sperm selection is based on HA-binding. While some improvements were expected, it is equally as important to note that no detrimental effects to clinical outcomes were associated with HAB-sperm selection, even in the "unselected" patient group. Therefore, this technique presents a valid option for the treatment of patient groups commonly excluded from RCTs. For example, a recent study and two meta-analyses have reported that male partners of couples experiencing recurrent pregnancy loss (RPL) demonstrate significantly higher sperm ROS concentrations and altered gene expression profiles [77], and increased levels of sperm DNA fragmentation [78,79], suggesting that high frequencies of DNA damage may be beyond the repair capabilities of the oocyte, and consequent chromosomal anomalies may be incompatible with post-implantation development and viability. Such patients are increasingly referred for assisted conception techniques despite an ability to naturally achieve pregnancy, while being underrepresented in embryological research [80].

Therefore, considering the apparent reduction in miscarriage following HAB-sperm selection, despite a lack of improvement in live birth rates, the application of HAB-sperm selection in this patient group could reduce the time to live birth per patient, rather than live birth rate per intention to treat as is commonly reported. This could provide real-life benefits to RPL patients, potentially reducing the number of distressing miscarriages while reducing the time to achieving a healthy live birth. This represents a promising area of future research, as it is apparent the current methodology is failing to realise any potentially relevant benefits to specific patient groups.

### 4.5. Limitations of Included Studies

The included studies claim to have focused on an "unselected" patient group, despite the inclusion of female factor infertility and the exclusion of male factor. Recently, a growing body of evidence has emerged suggesting that ICSI application to unexplained infertility populations is not only less effective compared with conventional IVF, but may be detrimental to clinical outcomes such as CPRs and LBRs [81], while also introducing an unknown risk of congenital abnormalities to the offspring. Therefore, it is not unreasonable to suggest that as the included studies largely focused on ICSI cycles in "unexplained infertility" patients, the benefits of HAB-sperm selection cannot fully be realised by such a study design. Given the above reports, HA-supplemented ICSI would likely only benefit patients traditionally referred for ICSI cycles, e.g., teratozoospermia, asthenozoospermia, and oligozoospermia. However, the most common exclusion criteria of the included studies were related to such conditions, excluding male factor infertility including sperm concentration of <10,000–1 million/mL [26,28,45,46,49], or <5% motility [28,46], patients routinely referred for ICSI.

This is supported by small-scale studies assessing only male factor infertility patients, including teratozoospermia and oligoasthenozoospermia [74,82], who not only reported a significant increase in CPRs in the PICSI treatment group compared with ICSI ($p = 0.009$), but also a more pronounced effect in the teratozoospermia patients, with PICSI serving

as an independent factor associated with obtaining at least one good-quality embryo in couples with severe teratozoospermia [82]. Furthermore, retrospective analysis of patients with high DFI also concluded that patients benefitted from some sort of intervention, e.g., TESE/TESA, PICSI, and IMSI [83], with LBRs comparable to those of patients with low DFI. Moreover, the included studies have essentially treated female factor infertility by not excluding both tubal and uterine factor infertility associated with poorer clinical outcomes, which cannot be improved by ICSI insemination. As emerging evidence confirms that a blanket approach to ICSI application is ineffective, it is unreasonable to assume that ICSI supplemented with HA-sperm selection would improve clinical outcomes in these patient groups.

Therefore, further study should actively research the impact of HAB-sperm selection either in conventional IVF cycles of unexplained infertility patients, for the treatment of recurrent pregnancy loss, or in ICSI cycles of patients suffering from male infertility. This would avoid the erroneous dismissal of the technique as ineffective despite real-life benefits to specific patient groups.

More recently, a case report has emerged reporting a healthy live birth and consecutive ongoing pregnancy in a case of severe globozoospermia treated with PICSI, following failed ICSI cycles [84], suggesting that HAB-sperm selection may be beneficial in cases of severe male factor where conventional treatments have failed. However, this singular case report is insufficient to draw robust conclusions on the impact of HAB-sperm selection on specific male infertility factors, highlighting an urgent need for not only further research, but also for the development of a better study design to ensure all patients who may benefit are not only identified, but included.

### 4.6. HAB Score as a Screening Tool

Conventional semen parameters are poor predictors of fertilisation in vivo or in vitro, and therefore the introduction of additional sperm parameters may assist in combatting the overuse of ICSI, by an enhanced ability to identify male patients eligible for IVF or ICSI insemination [85]. In an effort to tackle this lack of prognostic value, many studies have sought to develop screening procedures, which are largely focused on the measurement of DNA fragmentation via a DNA Fragmentation Index (DFI), with several meta-analyses reporting a significant correlation between DFI and clinical outcomes [78,86,87]. However, not only is there a lack of standardisation of measurement procedures, e.g., TUNEL, SCSA, Comet, etc., but these techniques require expensive equipment and are time-consuming. Moreover, the efficacy of measurement varies between techniques [86,88,89]. However, the materials required for HA-binding are readily available in all ART clinics, relatively inexpensive, and more importantly, HA-binding is associated with decreased levels of DNA fragmentation [26,28,90].

While the specificity of HAB scoring may not yet be optimised, this provides a cheap, effective, and quick measure of sperm fertilising ability via conventional IVF, and may therefore be used as a screening process prior to the selection of treatment modality and patient referral, while also providing key information of where defects may have arisen in cases resulting in fertilisation failure.

### 4.7. Future Approaches by the Clinic and the HFEA

The HFEA has understandably decided to promote a more transparent and honest approach to the utilisation of add-ons following numerous reports of promotion of treatments to improve profitability rather than patient care [91,92]. This led to the introduction of the "Traffic Light System". However, due to the regulatory scrutiny and financial burden of undertaking large studies [93], there is a paucity of high-quality RCTs reporting benefits of any such treatment, and thus far no add-ons have been given the green light for routine practice. While this does not represent a ban on the use of this technique, and may create a more ethical approach and reduce the potential financial exploitation of desperate patients, it also poses a massive obstacle to treatment innovation and the development of individu-

alised care, as for patients with specific infertility aetiologies, e.g., recurrent pregnancy loss and severe teratozoospermia, such treatment supplementation may represent a "last resort" to achieve and maintain a healthy pregnancy, however they will not be routinely offered. Moreover, these patients are routinely excluded from RCTs, and therefore the efficacy of such add-ons in the treatment of specific infertility phenotypes will never be realised by this "evidence-based medicine" approach.

As demonstrated by the included studies, RCTs are commonly designed to assess the use of such treatments in an empirical manner, studying their effects on disease presentation, e.g., unexplained infertility, rather than a specific disease pathology relevant to the scientific basis of the technique itself, in this case, specifically male factor infertility and poor sperm quality. This blanket application of emerging techniques to an unselected population may produce negative effects in some patients while masking the benefits felt by specific sub-populations, leading to the dismissal of treatments as ineffective despite real benefits felt by individual patients. This is the case with HA-sperm selection, as not only were patients predicted to benefit most from the treatment often excluded, but additional confounders were included, particularly both uterine and tubal factor infertility known to affect clinical outcomes but also not routinely referred for treatment by conventional ICSI, the chosen control in most studies. It is noteworthy, however, that despite the limitations of this RCT study design, significant improvements are still observed in the incidence of pregnancy loss.

Most included studies focused on conventional and HA-ICSI. Despite the current requirements of the HFEA, ICSI itself has never been subjected to the same scrutiny before its implementation for routine use [85]. Moreover, ICSI application in cases of non-male factor infertility is continuously rising [94]. While this is "justified" by a potential reduction in the risk of fertilisation failure [95], a wealth of data have reported that ICSI provides no real benefit to non-male factor patients [96], while some report that ICSI application to normozoospermic men may reduce the rates of clinical pregnancy and live birth [81]. Therefore, as well as an urgent need to understand the long-term health outcomes of ICSI conceived offspring [97], this widespread, blanket application of ICSI may reduce treatment effectiveness in individual cases. It is not therefore surprising that significant improvements to outcomes are not realised when HAB-sperm selection is applied to ICSI cycles in unselected populations compared with IVF.

This review has demonstrated that not only is HAB-sperm selection safe without any detrimental effects on clinical outcomes, but it can also reduce the incidence of hugely distressing miscarriages, which could in turn lead to improved live birth rates while reducing the financial burden of pregnancy loss aftercare or further ART cycles. Therefore, the red light assigned to PICSI by the HFEA is rather puzzling, as while the effectiveness of PICSI to improve live birth is currently unproven, no safety concerns have arisen in any published studies. Moreover, this assignment is likely to reduce the number of patients and clinical teams willing to utilise PICSI or HAB-sperm selection, despite potential real-life benefits, and may reduce the trust between patients and the clinicians who recommend its application. It is also important to note that despite its widespread use, ICSI itself has never been assigned a green light by the HFEA.

In order to combat this lack of supporting evidence, while ensuring patients have proper access to potentially beneficial treatment, it has been argued that HAB-sperm selection should not be considered an add-on, and instead introduced into routine practice [37]. This would not only allow specific patient sub-groups to reap the benefits of the technique, but via the collection of individualised patient data (IPD) would allow the HFEA to monitor the clinical outcomes of all UK cycles to produce a comprehensive report on the effects of HA-binding and sperm selection. Moreover, this IPD strategy ensures that specific patient groups will not be erroneously excluded from RCTs, resulting in premature dismissal of the treatment efficacy, while allowing for the development of individualised treatment approaches. An important example of this published during the final stages of proofing this manuscript, is a highly relevant paper, specifically pertaining to the previously published clinical HABSelect trial by Miller et al. (2019) [31], asking what effects HAB select ahead

of ICSI had on clinical outcomes. In brief, the findings were that older women who were randomised to the experimental arm of this trial (i.e., sperm selected through HA binding) had live birth rates not significantly different from that of younger women [98]. The authors postulate that this is probably due to "better avoidance" of sperm that had DNA damage. They also pointed out that although HABSelect was a prospective RCT, the group studied was from the recruitment cohort for retrospective analysis, and therefore did not have all the benefits of randomiation. They postulated that DNA damage in the sperm (reflected by lower hyaluronic acid binding) contributed to the depression of all gestational outcomes including live birth rates. They suggested that the interventional avoidance of defective sperm is the best to explain why the younger and older age groups were not different.

Currently, clinical and research staff in the UK must report all treatment outcomes to the HFEA, and therefore an IPD approach presents an opportunity to collect data on treatment effectiveness without the need for expensive, long-winded trials which may produce further inconclusive results. This would also assist in the identification of specific patient sub-groups who benefit from this intervention, and the development of clinical guidelines based on real-life practice.

Moreover, HA-binding is cost-effective and not detrimental to embryo development as HA is easily metabolised by the oocyte [20,22], and can be used in viscous media, e.g., SpermSlow, as a substitute for PVP during ICSI insemination, as studies have reported similar effectiveness of HA media when compared to PVP, without the associated impairment to embryo development [20,29]. This would minimise the changes required to current standard operating practices and the requirement of additional staff training.

*4.8. Study Limitations*

The conclusions made in this study are only as robust as the source studies. Considering the inconsistent reporting, exclusion and inclusion criteria, and methodological variations between studies, the results of this study are likely impacted by low study quality. However, this further highlights the need for more high-quality and robust research given the lack of understanding surrounding HA-binding sperm selection.

**5. Conclusions**

Optimal membrane structure and functionality of sperm is dependent on developmental and maturational events in spermiogenesis [9,10,14,16]. Mature sperm have receptors capable of binding to HA and the zona pellucida, and these sperm have been shown to have normal morphology and reduced levels of chromosomal aneuploidy and fragmented DNA. Unbound spermatozoa were shown to have lower levels of receptors for HA as well as higher levels of chromosomal aneuploidy and DNA fragmentation. Just as the zona plays a role in the selection of these mature sperm in conventional IVF insemination, HA-binding may be used as a selection tool for functional sperm in ICSI.

While there is little evidence to suggest HAB-sperm selection improves fertilisation or live birth rates, this technique reportedly improves the incidence of pregnancy loss following ART in an unselected infertility population. However, there is a paucity of data regarding other specific patient groups who would likely benefit from the implementation of this method, e.g., in cases of recurrent pregnancy loss or teratozoospermia. Furthermore, there is no evidence of any adverse effects of HAB-sperm selection on clinical outcomes, making the HFEA's decision to assign the technique a red light, and recommendations against the routine application of the method, rather puzzling.

These issues can be addressed in two ways: future RCTs should focus on these specific patient groups in order to ascertain if any real benefit is felt by these patients, or the HFEA should remove the "add-on" classification of HA-sperm selection and implement its routine use, in order to collect individualised patient data and better identify which patients, if any, will receive the most benefit from this treatment, and to fully understand the impact of this technique on all clinical outcomes.

**Author Contributions:** Conceptualization, T.M.; methodology, R.N.D. and T.M.; software, R.N.D. and T.M.; validation, R.N.D., T.M. and D.K.G.; formal analysis, R.N.D.; writing—original draft preparation, R.N.D.; writing—review and editing, T.M. and D.K.G.; supervision, T.M. All authors have read and agreed to the published version of the manuscript.

**Funding:** This research received no external funding.

**Institutional Review Board Statement:** Not applicable.

**Informed Consent Statement:** Not applicable.

**Data Availability Statement:** Not applicable.

**Acknowledgments:** Christopher Barratt and Vanessa Kay, Course Directors for the MSc in Human Clinical Embryology and Assisted Conception, Department of Medicine, University of Dundee. Lara Girod for contribution to the pictures in the graphical abstract, adapted from her MSc dissertation (University of Dundee, 2019). David Miller for the use of Figure 1 in the introduction.

**Conflicts of Interest:** The authors declare no conflict of interest.

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
