# Peer review of "The Efficacy of Hyaluronic Acid Binding (HAB) in the Treatment of Male Infertility: A Systematic Review of the Literature"

_2673-8856, doi:10.3390/dna2030011_

Round 1

Reviewer 1 Report

I would like to say this is a well conducted systematic review. However the conclusions are pretty similar to the earlier study in 2015 by Dr Beck-Fruchter. Please elaborate on the reasons of poor sperm quality associated with low hyaluron binding scores. 

Author Response

Response to Reviewer 1:

Thank you for your review and your comments. The conclusion would definitely benefit from a few lines regarding the maturity and ultrastructure of sperm in relation to binding. We have not included in the main text the scientific background regarding creatine kinase and chaperone proteins during spermiogenesis which explains synthesis of HA receptors, because this is a sizeable amount of information that would cloud the true purpose of the paper. However, this is information that is vital to include in some measure as it further enhances the benefits of sperm selection for ICSI especially in individualised cases. After much consideration (and several re-writes) a few lines have been added to the conclusion, which talks about sperm quality and binding, but also makes a stronger rationale for the outcome of the paper. We hope you find that this suffices.

Reviewer 2 Report

In the present manuscript written by Ni Dhuifin et al., the authors described the efficacy of Hyaluronic Acid Binding in the treatment of male infertility. The conducted review seem to be interesting. Research in this area is still inconclusive. In view of the increasing scale of infertility in recent years, it is important to search for new potential progonostatic factors in this area.

 The manuscript is well-written and interesting, the authors also noted the limitations of the research. However, below there are points, which should be taken into account:

 1. The authors should provide the explanation of all abbreviations in the manuscript (e.g. IVF, ART, ROS), where they are first used.

2. There should be a colon instead of the semicolon on line 123, 131, 162.

3. The information that the searches were conducted by Ní Dhuifin and Moodley (line 147, 168) should be in author contribution section.

4. Authors should consider whether it is worth preparing a diagram that summarizes the clinical use of HAB-sperm selection, which would undoubtedly enrich the manuscript.

Author Response

Reply to Reviewer 2:

Thank you for your insightful comments.

Reply to point 1:

The document was scanned and explanations for missing abbreviations have now been included throughout.

Reply to point 2:

Colons have replaced semi-colons in the requested lines. Please note that after other editing the line numbers have changed.

Reply to point 3:

The lines regarding author contributions have been removed and included in the appropriate sections.

Reply to point 4:

Thank you for this comment and we agree. A diagram has been inserted in the introduction which indeed adds to the explanation of the clinical use of HA-binding.